# Incidence, risk factors, and control of Rabies in Ethiopia: A systematic review and meta-analysis

**Belay Tafa Regassa**[1]*, **Wagi Tosisa**[1], **Daniel Eshetu**[2], **Andargachew Mulu**[3], **Gadissa Bedada Hundie**[4]

**1** Department of Medical Laboratory Sciences, College of Health Sciences and Referral Hospital, Ambo University, Ambo, Ethiopia, **2** Department of Microbiology, Yirgalem Hospital Medical College, Sidama Regional State, Yirgalem, Ethiopia, **3** Armauer Hansen Research Institute, Addis Ababa, Ethiopia, **4** Department of Microbiology, Immunology & Parasitology, St. Paul's Hospital Millennium Medical College, Addis Ababa, Ethiopia

* belaytf@gmail.com

## Abstract

### Background

Globally, rabies is found in several geographical areas, with tens of thousands of deaths annually, mainly in developing countries. However, though Ethiopia is highly endemic for rabies, the overall risk of rabies has not yet been estimated. Hence, this systematic review and meta-analysis aimed at estimating a pooled incidence rate of human exposure to suspected rabid animals as well as the incidence rates of rabies in humans and other domestic animals.

### Methods

Published articles search was systematically performed through PubMed, Scopus, Embase, and Web of Science databases to identify the available studies on rabies until October 2023. The Joanna Brigg's Institute (JBI) critical appraisal checklists were used for assessing the quality of the studies. The PRISMA 2020 guideline was followed. A qualitative synthesis was made describing the characteristics of the included studies. The quantitative synthesis was performed with a random effects model using Comprehensive Meta-Analysis (CMA) version 3.0 software. The Q statistic quantified by $I^2$ was used to check for heterogeneity among the included studies. To explain the source of heterogeneity, subgroup analysis was performed. Egger's regression test was used to evaluate publication biases. This study is registered with PROSPERO, CRD42023468791.

### Results

For this study, a total of 439 articles were retrieved; of which fifteen studies were included in the final review. The annual pooled incidence rate of human exposure to suspected rabid animals was 33.65 (95% CI: 31.82 to 35.49) per 100,000 humans. The suspected rabies deaths in humans were also estimated to be 4.57 (95% CI: 2.93 to 6.21) per one

**Data availability statement:** All relevant data are within the paper and its supporting information files.

**Funding:** The author(s) received no specific funding for this work.

**Competing interests:** The authors have declared that no competing interests exist.

million humans annually. In both cases, considerable heterogeneities were presented across the included studies, and obvious publication biases were detected using Egger's regression test. Among animals, the highest combined estimate per 100,000 population was recorded in dogs, 120.99 (95% CI: 46.29 to 195.69), followed by equines and cattle, with pooled incidence rates of 19.57 (95% CI: −1.85 to 40.98) and 18.08 (95% CI: 1 to 35.15), respectively. It was also described that human exposure to rabid animals and human rabies deaths were more common among children.

## Conclusions

The current study showed a high pooled incidence rate of human exposure to rabid animals. Significant overall incidence rates of rabies in humans and animals were also indicated. Therefore, strengthening intersectoral and transdisciplinary collaborations through one health approach are key components for rabies prevention and control.

## Author summary

Rabies virus infects all mammals with a fatality rate of nearly 100%. More than 99% of human rabies cases are transmitted via dog bites. Becoming one of the neglected tropical diseases, rabies occurs in several geographical areas. In Ethiopia, rabies vaccination coverage among dogs is far below the optimal 70%. This is due to the limited anti-rabies vaccination, lack of awareness among dog owners, and a high number of stray dogs. However, the overall risk of rabies has not yet been estimated in the country. Information on rabies incidence is important for notifying the population at risk and for planning effective prevention, control, and elimination strategies. We performed a systematic review and meta-analysis of published studies in Ethiopia. The annual pooled incidence rate of human exposure to suspected rabid animals was 33.65 per 100,000 humans. The suspected rabies deaths in humans were also estimated to be 4.57 per one million humans annually. Among animals, the highest combined estimate per 100,000 populations was recorded in dogs, 120.99, followed by equines and cattle, with pooled incidence rates of 19.57 and 18.08, respectively. We also found that human exposure to rabid animals and human rabies deaths were more common among children.

## Introduction

Rabies virus infects all mammals, including humans, with a fatality rate of nearly 100%. It occurs in over 150 countries, and almost half of the global population is at risk of contracting rabies living in canine rabies-endemic areas. It causes tens of thousands of deaths annually, becoming one of the neglected zoonoses, mainly in Asia and Africa [1,2]. More than 99% of human rabies cases are transmitted via dogs [1], and human-to-human transmission is possible [3]. Rabies impacts the budget of public health sectors, local communities, and livestock economies, particularly in poorer regions of the world. Globally, canine rabies causes 59,000 human deaths, 3.7 million disability-adjusted life years (DALYs), and over 8.5 billion USD in annual economic losses [4].

Rabies requires coordinated activities at national, regional, and global levels. It is included in the WHO's 2021–2030 Roadmap for the Global Control of Neglected Tropical Diseases [2]. While canine rabies has been eliminated in some countries, it remains prevalent in over

80 countries, especially in low- and middle-income countries like Ethiopia [1,4]. The high endemicity is attributed to the large dog population and poor management [5]. According to the CDC, Ethiopia experiences one of the highest rabies death rates globally, with 2,700 deaths annually [6]. The disease also causes approximately 194,000 DALYs and 97,000 exposed persons annually, requiring an average of 2 million USD in treatment costs [7]. However, the true number of rabies deaths is unknown due to underreporting and limited rabies diagnostic laboratories [6].

Rabies is a 100% preventable disease in both animals and humans, with dog vaccination being the most cost-effective strategy [2,8]. However, rabies vaccination coverage in Ethiopia is low, below the optimal 70% required to prevent canine rabies transmission [6]. In Addis Ababa, only 27% of the dogs inflicting bites are vaccinated, and 70.3% of them are laboratory-confirmed rabid, which is alarming [9]. This could be due to limited anti-rabies vaccination, lack of awareness among dog owners, and high stray dog populations [6,10].

The World Health Organization recommends pre-exposure prophylaxis for those with increased risk, for example, veterinarians, laboratory workers, and animal handlers. In cases where individuals are exposed to a rabid animal, immediate vaccination and the administration of rabies immunoglobulin for high-risk exposures are indicated [2]. However, few places in Ethiopia offer life-saving human rabies post-exposure prophylaxis (PEP). Poor awareness about rabies prevention also prevents people from seeking medical help when bitten by dogs [6].

The annual livestock losses caused by rabies place a large societal and economic burden on the country, although the total number of animal rabies cases in Ethiopia is unknown. In humans and other animals, the overall prevalence of rabies was estimated to be 32% in Ethiopia [11]. Information on the overall rabies incidence rates is important for notifying the population at risk of the disease and for planning effective prevention, control, and elimination strategies. However, to the best of our knowledge, the pooled incidence rates of rabies in human and animal populations have not yet been estimated in the country. The absence of such summarized data on the incidence of rabies, both in humans and animals, might be among the major problems that cause negligent commitment to the control of rabies, as still Ethiopia is in the early stages of rabies control [12].

Hence, this systematic review and meta-analysis aimed at providing an overall estimate of the incidence rate of human exposure to suspected rabid animals as well as the incidence rates of rabies in humans and other domestic animals. Furthermore, it explored the associated risk factors of human rabies, its exposures, and control measures based on the published literature.

## Methods

### Databases and search strategies

The review protocol has been registered with the registration number CRD42023468791 in the International Prospective Register of Systematic Reviews (PROSPERO). A search was comprehensively performed through PubMed, Scopus, Embase, and Web of Science databases. Keywords, medical subject headings (MeSH), and related terms were included in our search strategies. The search terms and strings for databases were provided as supplementary information (S1 Information). We have also searched for grey literature through grey literature reports, worldwide science, and Ethiopian universities as well as research institute repositories. Further, we searched articles from reference lists of included studies and related systematic reviews and meta-analyses through African Journal Online (AJOL) and Google Scholar.

## Inclusion and exclusion criteria

The inclusion and exclusion criteria were predefined and used to screen titles and abstracts and evaluate full texts for eligibility. Observational studies, including cross-sectional and cohort (both retrospective and prospective) studies, were included. Studies addressing the incidence, risk factors, and/or control of rabies in humans and/or animals from Ethiopia were considered. Clinically or laboratory-confirmed cases of rabies as well as a history of dog or animal bites as sources of rabies exposure were considered. English-language literature and studies conducted in Ethiopia until October 6, 2023, from the time of the inception of each database, were included. On the other hand, all original articles outside of Ethiopia, review papers, books, letters, brief reports, case reports, and poster presentations were excluded during the screening of titles and abstracts. Articles with unrelated outcome measures and those with missing or insufficient outcome(s) were also excluded during the screening of the full text.

## Study selection and quality assessment

Firstly, all retrieved studies were imported to the Endnote X9 citation manager. Duplicated studies were automatically removed from the Endnote. According to the inclusion and exclusion criteria, two reviewers independently screened all articles retrieved by the search strategy by title and abstract for eligibility. In the process, any discrepancies between the reviewers were discussed and resolved jointly to reach a consensus, and, if necessary, the full text was accessed for further clarification. The full texts of the eligible articles were then retrieved and assessed by the two reviewers. Following the full-text review, two independent investigators assessed the quality of the studies using the Joanna Brigg's Institute (JBI) critical appraisal tools adapted for observational studies. In the case of discrepancies, the two reviewers discussed and resolved the issues. Otherwise, the decision of the third reviewer was accepted. Accordingly, studies with positive responses ("yes") for greater than or equal to half (≥50%) of the number of checklist items were graded as good quality articles and those with less than half (<50%) scores were considered poor quality articles.

## Data extraction and management

Data extraction was performed on an Excel spreadsheet and included the following: study first author and year of publication, study design, study setting (community, health facility, or both), study population (humans, animals, or both), number of new cases during follow-up, the total number of the at-risk population, risk factors, control measures, and administrative regions or cities of Ethiopia. Furthermore, person/animal-year and annual incidence rates of rabies/rabies exposures were computed from the available data and included in the data extraction format (S2 Information). Extractions were done by two independent reviewers, and if disagreements happened during the data extraction process, they were resolved through joint discussions between the reviewers. The data were used only once if the results were published multiple times.

## Strategy for data synthesis

The PRISMA 2020 guideline was followed for a systematic review and meta-analysis [13]. A flow diagram was used to illustrate the literature search and article selection processes (Fig 1). A table was also used to provide an overview of the characteristics of the included articles. The outcomes of the study were the combined incidences of the conditions of interest among the target population categories (incidences of rabies exposure or rabies in humans, incidence of rabies in dogs, and other animal species). A qualitative synthesis was

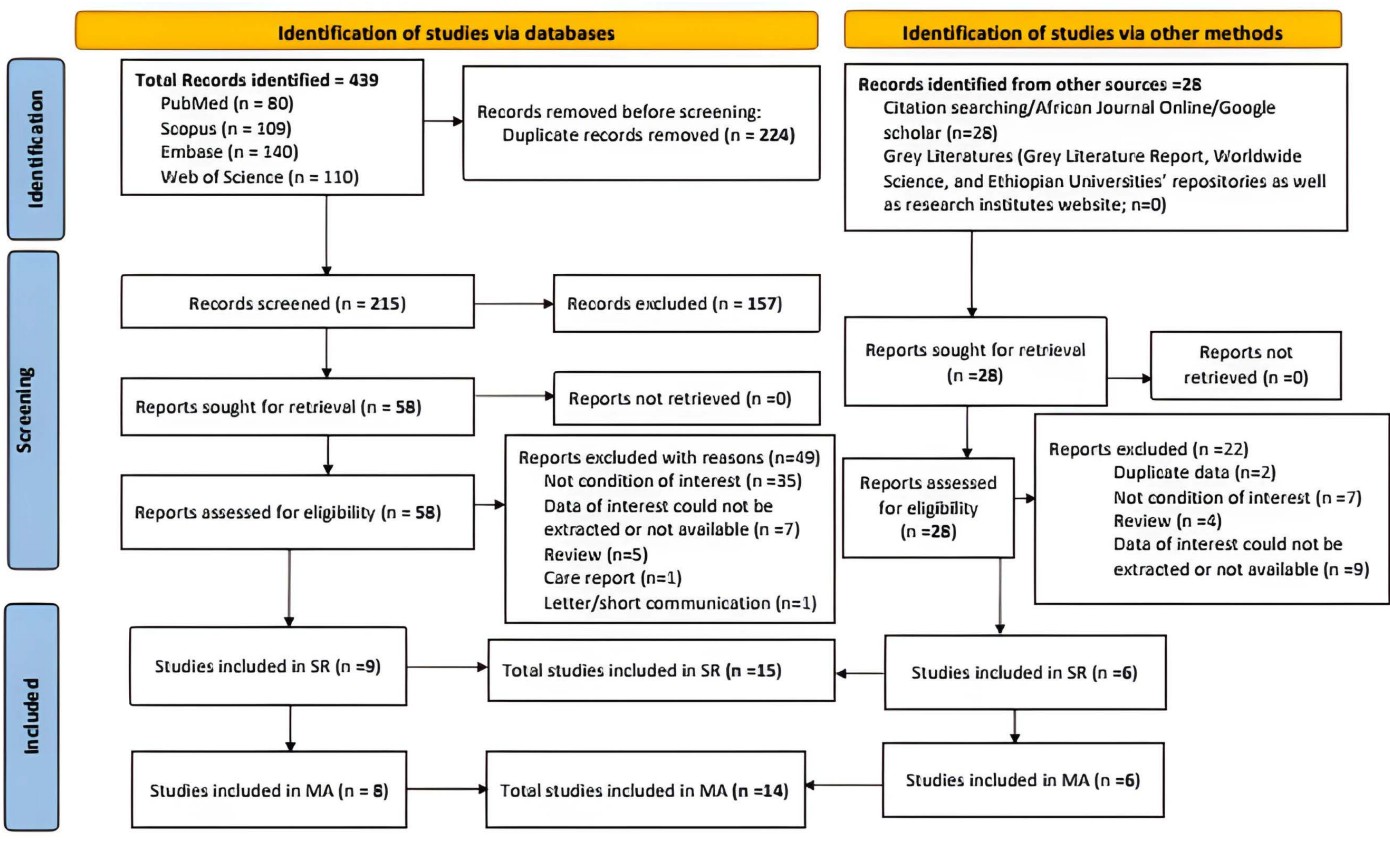

**Fig 1. PRISMA 2020 Flow diagram for literature search and selection.**

made describing the characteristics of the included studies. The quantitative synthesis was performed with a random effects model. Comprehensive Meta-Analysis (CMA) Version 3.0 statistical software was used for meta-analysis so that an overall summary effect estimate of the incidence across studies was determined. Further, we checked for heterogeneity across studies and also performed subgroup analysis to identify the sources of heterogeneity. Heterogeneity was evaluated by the chi-square test on Cochrane's Q statistic, which was quantified by $I^2$ values. The $I^2$ statistic estimates the percentage of total variation across studies due to true between-study differences rather than chance. Sensitivity analyses were conducted to check the stability of the summary estimate, and publication biases were evaluated using Egger's regression test.

## Results

### Review processes and findings

Four databases were accessed and a total of 439 articles (PubMed = 80; Scopus = 109; Embase = 140; and Web of Science = 110) were found without limiting when publications started until October 6, 2023. After removing the duplicates (n = 224) using the Endnote reference manager, 215 records were kept. After screening for titles and abstracts, 157 irrelevant studies were removed, and 58 full-text papers were retrieved and assessed for eligibility. Of the 58 full-text papers, because of different reasons, 49 articles were excluded, and 9 were found relevant. In addition to the databases, records were also sought from other sources such as free

web search engines (African Journal Online and Google Scholar) and Grey Literature, and as a result, we obtained 28 full-text records. Assessing for eligibility, 22 records were excluded with reasons, while 6 were found relevant for the review. According to the quality assessment tool, all the included studies had good quality scores. Finally, 15 and 14 relevant articles were included in the systematic review and meta-analysis, respectively (Fig 1).

## Characteristics of the included studies

The study records were reported from four regional states and one city administration in Ethiopia. By region, 5 studies (33.3%) were from Oromia [7,14–17], 4 (26.7%) were from Amhara [18–21], 3 (20.0%) from Tigray [22–24], and one (6.7%) study was multi-regional [25] reporting aggregate data from three regions (Afar, Amhara and Tigray), while 2 (13.3%) studies were from Addis Ababa City Administration [26,27]. The majority, 80% (12/15), of included studies employed a cross-sectional study design [7,17,20,22,24,27] where almost all of them used retrospective timing frames. The target study population was humans in 60% (9/15) of the included studies [7,15,16,20,24,27], while it was 20% (3/15) each for animals [14,18,26] and both humans and animals [17,19,25]. Regarding the settings used for conducting the studies, health facilities, animal clinics, and the community were utilized. More than half (53.3%) of the studies were conducted in or utilized records of health facilities only [15,16,20,24,27]. One or more outcomes were reported from each of the included studies, giving a total frequency of 29 outcomes: nine studies reported the incidence of human exposure to suspected rabid animals; three outcomes, such as the incidence of suspected rabies in humans and rabies in animals, and risk factors for exposures or rabies, were documented in 6 (20.7%) studies each; and the other 2 (6.9%) studies reported prevention/control measures taken to prevent rabies (Table 1).

## Number of studies/datasets included in systematic review and meta-analysis

In total, 15 studies were considered for the systematic review [7,14–27]. Of these, 14 (93.3%) were included in the meta-analysis, where different numbers of studies and datasets were used for the different outcomes reported. Nine studies with 23 datasets were used in the meta-analysis for the incidence of human exposure to suspected rabid animals [7,16,19,24,27], while 6 studies with 14 datasets were included for the incidence of suspected rabies in humans [7,17,19,22,25,27]. For the incidence of suspected rabies by animal types such as goats [17,19], cats [17,26], equines [17,19,25], dogs [17,19,25,26], and cattle [14,17,19,25], 2 to 5 studies were included in the meta-analysis. Since the incidence of suspected rabies in camels was reported by one study [25], with a single dataset, it was not considered for meta-analysis. Moreover, 6 and 2 studies reporting the risk factors for rabies exposure or rabies deaths [15,16,19,20,24,25] and control measures [16,24] were included in the systematic review, respectively (Table 2).

## Incidences of human exposure to rabid animals and human rabies from included studies

Overall, 11, 294 human exposures to suspected rabid animals were reported from the included studies. The reported incidence rates of human exposures ranged from zero in 2015 [27], to 422 per 100,000 humans in 2013 [22]. In general, the highest incidences of human exposure to suspected rabid animals were reported in the Tigray region, ranging from 278 to 422 per 100,000 populations [22]. Relatively, maximum incidences of exposures of 108.64 per 100,000 [7] and 200.64 per 100,000 [16] were also reported from Oromia regional state. The smallest estimates of incidences of exposures were reported from Addis Ababa and North Gondor,

**Table 1. Profiles of studies included in the systematic review and meta-analysis.**

| Study profiles | | Included studies Number (%) |
|---|---|---|
| Publication year | 2013 | 1 (6.7) |
| | 2014 | 1 (6.7) |
| | 2015 | 2 (13.3) |
| | 2016 | 2 (13.3) |
| | 2017 | 1 (6.7) |
| | 2018 | 3 (20.0) |
| | 2019 | 2 (13.3) |
| | 2022 | 3 (20.0) |
| Study design | Cross-sectional | 12 (80.0) |
| | Cohort | 3 (20.0) |
| Study setting | Health facility only | 8 (53.3) |
| | Community only | 3 (20.0) |
| | Health facilities and community | 1 (6.7) |
| | Health facility and animal clinic | 2 (13.3) |
| | Animal clinic only | 1 (6.7) |
| Target population | Humans | 9 (60.0) |
| | Animals | 3 (20.0) |
| | Both | 3 (20.0) |
| Region | Oromia | 5 (33.3) |
| | Amhara | 4 (26.7) |
| | Tigray | 3 (20.0) |
| | Addis Ababa | 2 (13.3) |
| | Multi-regional (Afar, Amhara, Tigray) | 1 (6.7) |
| Outcomes reported* | Incidence of human exposure to suspected rabid animals | 9 (30.0) |
| | Incidence of suspected rabies in humans | 6 (20.7) |
| | Incidence of suspected rabies in animals | 6 (20.7) |
| | Risk factors for rabies exposure or rabies | 6 (20.7) |
| | Prevention/control measures for rabies | 2 (6.9) |

A health facility refers to a hospital or health center.

*For a single study, one or more outcome measures were reported.

ranging from 0 to 1.97 [27] and 1.27 to 4.6 [20], respectively. However, higher incidences of exposures, 12.41 and 24.8 per 100,000, were recorded in the datasets from Addis Ababa [27], which were higher than the reports from Bahir Dar and other zones [21] (Table 3).

Regarding suspected rabies in humans, a total of 303 deaths were reported from different corners of Ethiopia, as indicated in the included studies. The annual incidence rate of human rabies recently reported from the Buno Bedele zone of Oromia was 4.19 per 100,000 population [17]. A multi-regional record from three regions, Afar, Amhara, and Tigray, also reported that the incidence rate of rabies in humans was 4.13 per 100,000 populations [25]. Annually, incidence rates of human rabies ranging from 1 to 2.94 per 100,000 populations were also reported in other studies [7,19,22]. On the other hand, rare occurrences (≤1 per million) of suspected human rabies were recorded in some datasets from Addis Ababa [27] (Table 3).

## Incidence rates of rabies in animals from included studies

Some studies reported the incidence rates of rabies in various domestic animals (Table 4). Among animals, the first and second highest estimates of suspected incidence per 100,000

**Table 2. Studies and datasets included in systematic review and meta-analysis by outcome.**

| The outcome measured or reported by studies | | Number of studies included in SR | Studies included in the MA | |
|---|---|---|---|---|
| | | | Number of Studies | Number of datasets used |
| Incidence of human exposure to suspected rabid animals | | 9 | 9 | 23 |
| Incidence of suspected rabies in humans | | 6 | 6 | 14 |
| Incidence of suspected rabies by animal type | cattle | 5 | 5 | 5 |
| | dogs | 4 | 4 | 4 |
| | cats | 2 | 2 | 2 |
| | equines | 3 | 3 | 3 |
| | goats | 2 | 2 | 2 |
| | camels | 1 | NA | – |
| Risk factors for rabies exposure or rabies | | 6 | – | – |
| Prevention/control measures | | 2 | – | – |

NA = Not applicable

animals, 1880.19 [14] and 911.16 [18], were reported in cattle, followed by that of dogs, 412.83 per 100,000 [19]. The smallest incidence (0.3 per 100,000) was also reported in cattle [25]. Relatively, other studies reported smaller estimates of rabies incidence in animals. All the included studies reported the incidence rates of suspected rabies, except one study [26] which reported the incidence of both suspected and laboratory-confirmed rabies in dogs and cats. Differences in the incidence rates per 100,000 populations were observed between suspected and confirmed cases of dogs (92.8 versus 70.2) and cats (5.7 versus 3.2); higher estimates were recorded for suspected animals [26]. Only one study reported the incidence of rabies in camels, 0.69 per 100,000 populations [25].

## Risk factors of exposure to rabid animals or rabies in humans, and control measures

A few numbers of socio-demographic features of humans were reported as risk factors for the occurrence of exposure to suspected rabid animals or rabies deaths in humans [15,16,19,20,24,25]. Studies from Amhara and Tigray regional states reported that male sex was more associated with human exposure to suspected rabid animals [20,24].

Regarding the age of exposed humans to suspected rabid animals, two separate studies from Jimma and Arsi, Oromia, reported that children < 15 years of age were much more likely exposed [15,16]. On the contrary, a study from the northwestern Tigray region reported that adult populations aged ≥15 years were most exposed [24].

As to the residence, rural was recorded as the major factor for exposure to rabid animals from Amhara and Oromia [15,19]. However, another study from Amhara reported that urban residents were more affected than rural ones [20].

Concerning the risk factors for rabies mortality in humans, female sex and the age of 5–14 years were identified as risk factors [25]. Dog bites were the source of exposure for all rabies mortality, as indicated by one study [19].

From the records collected for review purposes, a limited number of studies reported what measures were taken once humans were exposed to suspected rabid animals or the status of vaccination of dogs/animals inflicting bites. One study from the west Arsi zone of Oromia reported that all the exposed individuals to rabid animals received nerve tissue vaccines, but only 1% of cases did not complete the vaccination course [16]. On the other side, a study from northwestern Tigray published information that all the dogs inflicting unprovoked bites were unvaccinated [24].

Table 3. **Incidence rates of human exposure to rabid animals and human rabies from included studies.**

| Outcome | Studies | Dataset | Annual incidence rate per 100,000 population | Region or City |
|---|---|---|---|---|
| Human exposure to suspected rabid animals | Beyene *et al.*, 2018 [7] | – | 108.64 | Oromia |
| | Gebru *et al.*, 2019 [23] | – | 40.00 | Tigray |
| | Jemberu *et al.*, 2013 [19] | – | 24.94 | Amhara |
| | Ramos *et al.*, 2015 [16] | – | 200.64 | Oromia |
| | Teklu *et al.*, 2017 [24] | 2012 data | 35.83 | Tigray |
| | | 2013 data | 62.96 | |
| | | 2014 data | 89.75 | |
| | | 2015 data | 73.1 | |
| | Yibrah & Damtie, 2015 [20] | 2011 data | 4.6 | Amhara |
| | | 2012 data | 2.61 | |
| | | 2013 data | 1.27 | |
| | Yizengaw *et al.*, 2018 [21] | 2015/16 data | 6.48 | Amhara |
| | | 2016/17 data | 7.54 | |
| | Ebuy *et al.*, 2019 [22] | 2012 data | 392 | Tigray |
| | | 2013 data | 422 | |
| | | 2014 data | 278 | |
| | | 2015 data | 308 | |
| | | 2016 data | 317 | |
| | Mesfin, 2022 [27] | 2015 data | 0 | Addis Ababa |
| | | 2016 data | 0.1 | |
| | | 2017 data | 1.97 | |
| | | 2018 data | 12.41 | |
| | | 2019 data | 24.8 | |
| Suspected rabies in humans | Beyene *et al.*, 2018 [7] | – | 2.8 | Oromia |
| | Jemberu *et al.*, 2013 [19] | – | 2.34 | Amhara |
| | Menghistu *et al.*, 2018 [25] | – | 4.13 | Multi-regional (Afar, Amhara, Tigray) |
| | Ebuy *et al.*, 2019 [22] | 2012 data | 2.94 | Tigray |
| | | 2013 data | 0.98 | |
| | | 2014 data | 1.11 | |
| | | 2015 data | 2.75 | |
| | | 2016 data | 1.68 | |
| | Mesfin, 2022 [27] | 2015 data | 0 | Addis Ababa |
| | | 2016 data | 0 | |
| | | 2017 data | 0.12 | |
| | | 2018 data | 0.02 | |
| | | 2019 data | 0.04 | |
| | Wakgari *et al.*, 2022 [17] | – | 4.19 | Oromia |

## Pooled incidence rate of human exposure to suspected rabid animals

The overall pooled incidence rate of human exposure to suspected rabid animals per 100,000 humans annually was 33.65 (95% CI: 31.82 to 35.49). Considerable heterogeneity was presented for the overall combined effect size (Q = 11222.628 (df = 22), $P < 0.001$; I-squared = 99.804%). Of note, the overall true effect size was greater than zero (Z-value = 35.92, $P < 0.001$) (Fig 2).

**Table 4. Incidence rates of suspected and confirmed rabies in animals from included studies.**

| Animal type | Studies | Annual incidence rate per 100,000 population |
|---|---|---|
| Cattle | Alemu et al., 2022 [18] | 911.16 |
| | Jemberu et al., 2013 [19] | 19.89 |
| | Jibat et al., 2016 [14] | 1880.19 |
| | Menghistu et al., 2018 [25] | 0.3 |
| | Wakgari et al., 2022 [17] | 17.5 |
| Dogs | Jemberu et al., 2013 [19] | 412.83 |
| | Menghistu et al., 2018 [25] | 15.28 |
| | Wakgari et al., 2022 [17] | 180 |
| | Kidane et al., 2016 [26] | 92.8 <br> 70.2$^{\Omega}$ |
| Cats | Wakgari et al., 2022 [17] | 22.8 |
| | Kidane et al., 2016 [26] | 5.7 <br> 3.2$^{\Omega}$ |
| Equines | Jemberu et al., 2013 [19] | 67.68 |
| | Menghistu et al., 2018 [25] | 1.62 |
| | Wakgari et al., 2022 [17] | 23.7 |
| Goats | Jemberu et al., 2013 [19] | 14.45 |
| | Wakgari et al., 2022 [17] | 3.7 |
| Camels | Menghistu et al., 2018 [25] | 0.69 |

$^{\Omega}$Laboratory confirmed cases; otherwise, suspected cases are based on clinical definition.

## Subgroup analysis of incidence rates of human exposure to rabid animals by regions

In the meta-analysis of the incidence rates of human exposures to suspected rabid animals, substantial heterogeneity was found among the effect sizes of the included individual studies/datasets. Subgroup analysis was performed by regions of the studies/datasets to explore if they were the source of heterogeneity. The highest pooled incidence rate of human exposure to suspected rabid animals per 100,000 populations was recorded in Oromia, 138.91 (95% CI: 129.79 to 148.03), followed by Tigray regional state, 83.32 (95% CI: 80.25 to 86.39). The smallest pooled incidence by region was from Amhara, 5.52 (95% CI: 3.26 to 7.79), while it was 7.58 (95% CI: 5.25 to 9.91) from Addis Ababa. Considerable heterogeneities were also seen from the subgroup analysis between regions, indicating that regions of the studies would be sources of heterogeneity (Fig 3).

## Sensitivity analysis and publication bias for incidence of human exposure to rabid animals

Performing the sensitivity analysis by removing one study or dataset at a time, the overall results were noticeably changed when two datasets (datasets of 2015 and 2016 each) from the study reported by Mesfin [27] were omitted one by one, giving the pooled incidence rates of 64.84 (95% CI: 60.97 to 68.70) and 63.61 (95% CI: 59.85 to 67.37) per 100,000, respectively. Otherwise, the results were comparable with the overall effect when none of the studies were removed (Fig 4). To check for publication bias, Egger's regression test was performed, and obvious publication bias was detected among the publications that reported the incidence of human exposure to suspected rabid animals (Intercept = 22.06 (95% CI: 18.13 to 26.00), P < 0.001).

## Incidence rate of human exposure to suspected rabid animals

| Study name | Statistics for each study | | | Rate and 95% CI |
|---|---|---|---|---|
| | Rate | Lower limit | Upper limit | |
| Beyene et al., 2018 | 0.001086449 | 0.000987700 | 0.001185197 | |
| Ebuy et al., 2019 (2012 data) | 0.003919991 | 0.003648185 | 0.004191797 | |
| Ebuy et al., 2019 (2013 data) | 0.004220008 | 0.003938457 | 0.004501558 | |
| Ebuy et al., 2019 (2014 data) | 0.002779998 | 0.002607523 | 0.002952474 | |
| Ebuy et al., 2019 (2015 data) | 0.003080000 | 0.002889768 | 0.003270233 | |
| Ebuy et al., 2019 (2016 data) | 0.003169999 | 0.002967566 | 0.003372432 | |
| Gebru et al., 2019 | 0.000399927 | 0.000359066 | 0.000440787 | |
| Jemberu et al., 2013 | 0.000249715 | 0.000163195 | 0.000336235 | |
| Mesfin, 2022 (2015 data) | 0.000000129 | -0.000000229 | 0.000000487 | |
| Mesfin, 2022 (2016 data) | 0.000000990 | 0.000000020 | 0.000001960 | |
| Mesfin, 2022 (2017 data) | 0.000019687 | 0.000015452 | 0.000023922 | |
| Mesfin, 2022 (2018 data) | 0.000124091 | 0.000113682 | 0.000134499 | |
| Mesfin, 2022 (2019 data) | 0.000248040 | 0.000233635 | 0.000262445 | |
| Ramos et al., 2015 | 0.002006363 | 0.001855894 | 0.002156832 | |
| Teklu et al., 2017 (2012 data) | 0.000358277 | 0.000316899 | 0.000399655 | |
| Teklu et al., 2017 (2013 data) | 0.000629606 | 0.000575229 | 0.000683983 | |
| Teklu et al., 2017 (2014 data) | 0.000897465 | 0.000833107 | 0.000961824 | |
| Teklu et al., 2017 (2015 data) | 0.000730987 | 0.000673906 | 0.000788067 | |
| Yibrah and Damtie, 2015 (2011 data) | 0.000046009 | 0.000038388 | 0.000053631 | |
| Yibrah and Damtie, 2015 (2012 data) | 0.000026116 | 0.000020429 | 0.000031803 | |
| Yibrah and Damtie, 2015 (2013 data) | 0.000012652 | 0.000008731 | 0.000016572 | |
| Yizengaw et al., 2018 (2015/2016 data) | 0.000064803 | 0.000058627 | 0.000070978 | |
| Yizengaw et al., 2018 (2016/2017 data) | 0.000075419 | 0.000068815 | 0.000082023 | |
| **Overall estimate** | 0.000336512 | 0.000318150 | 0.000354874 | |

-0.01 -0.01 0.00 0.01 0.01

Heterogeneity: Q-value = 11222.628 (df = 22), P-value = 0.000
I-squared = 99.804
Tau-squared = 0.000
Test of null: z-value = 35.920, p-value = 0.000

**Fig 2. Forest plot for the incidence rates of human exposure to suspected rabid animals [ 7,16,19,24,27].**

### Pooled incidence rate of suspected human rabies

The annual pooled incidence rate of suspected human rabies was 4.57 (95% CI: 2.93 to 6.21) per one million humans. It was indicated that the overall true effect size was greater than zero (Z-value = 5.461, $P < 0.001$). Considerable heterogeneity was presented for the overall combined effect size (Q = 295.141 (df = 13), $P < 0.001$; I-squared = 95.595%) (Fig 5).

Conducting sensitivity analysis, the overall results were not noticeably changed except in the case of removing one study reporting the incidence of human rabies by Menghistu and others [25] where the combined effect of the rest of the studies/datasets gave a very small pooled incidence of < 1 per one million (Fig 6). Obvious publication bias was also detected using Egger's regression test among the publications that reported the incidence of human rabies (Intercept ($B_o$) = 4.01505 (95% CI: 1.43272 to 6.59738), P = 0.0027).

### Pooled incidence rate of suspected rabies in animals

The annual incidence rates of suspected rabies were pooled for different domestic animals in Ethiopia. The highest combined effect per 100,000 population was recorded in dogs, 120.99

## Incidence rate of human exposure to suspected rabid animals by regions in Ethiopia

| Group by Region | Study name | Rate | Lower limit | Upper limit | Rate and 95% CI |
|---|---|---|---|---|---|
| Addis Ababa | Mesfin, 2022 (2015 data) | 0.000000129 | -0.000000229 | 0.000000487 | |
| Addis Ababa | Mesfin, 2022 (2016 data) | 0.000000990 | 0.000000020 | 0.000001960 | |
| Addis Ababa | Mesfin, 2022 (2017 data) | 0.000019687 | 0.000015452 | 0.000023922 | |
| Addis Ababa | Mesfin, 2022 (2018 data) | 0.000124091 | 0.000113682 | 0.000134499 | |
| Addis Ababa | Mesfin, 2022 (2019 data) | 0.000248040 | 0.000233635 | 0.000262445 | |
| Addis Ababa | | 0.000075791 | 0.000052476 | 0.000099106 | |
| Amhara | Jemberu et al., 2013 | 0.000249715 | 0.000163195 | 0.000336235 | |
| Amhara | Yibrah and Damtie, 2015 (2011 data) | 0.000046009 | 0.000038388 | 0.000053631 | |
| Amhara | Yibrah and Damtie, 2015 (2012 data) | 0.000026116 | 0.000020429 | 0.000031803 | |
| Amhara | Yibrah and Damtie, 2015 (2013 data) | 0.000012652 | 0.000008731 | 0.000016572 | |
| Amhara | Yizengaw et al., 2018 (2015/2016 data) | 0.000064803 | 0.000058627 | 0.000070978 | |
| Amhara | Yizengaw et al., 2018 (2016/2017 data) | 0.000075419 | 0.000068815 | 0.000082023 | |
| Amhara | | 0.000055244 | 0.000032639 | 0.000077850 | |
| Oromia | Beyene et al., 2018 | 0.001086449 | 0.000987700 | 0.001185197 | |
| Oromia | Ramos et al., 2015 | 0.002006363 | 0.001855894 | 0.002156832 | |
| Oromia | | 0.001389141 | 0.001297911 | 0.001480370 | |
| Tigray | Ebuy et al., 2019 (2012 data) | 0.003919991 | 0.003648185 | 0.004191797 | |
| Tigray | Ebuy et al., 2019 (2013 data) | 0.004220008 | 0.003938457 | 0.004501558 | |
| Tigray | Ebuy et al., 2019 (2014 data) | 0.002779998 | 0.002607523 | 0.002952474 | |
| Tigray | Ebuy et al., 2019 (2015 data) | 0.003080000 | 0.002889768 | 0.003270233 | |
| Tigray | Ebuy et al., 2019 (2016 data) | 0.003169999 | 0.002967566 | 0.003372432 | |
| Tigray | Gebru et al., 2019 | 0.000399927 | 0.000359066 | 0.000440787 | |
| Tigray | Teklu et al., 2017 (2012 data) | 0.000358277 | 0.000316899 | 0.000399655 | |
| Tigray | Teklu et al., 2017 (2013 data) | 0.000629606 | 0.000575229 | 0.000683983 | |
| Tigray | Teklu et al., 2017 (2014 data) | 0.000897465 | 0.000833107 | 0.000961824 | |
| Tigray | Teklu et al., 2017 (2015 data) | 0.000730987 | 0.000673906 | 0.000788067 | |
| Tigray | | 0.000833213 | 0.000802495 | 0.000863931 | |

-0.01    -0.01    0.00    0.01    0.01

| Region | Test of null (2-Tail) | Heterogeneity statistics | | Tau-squared |
|---|---|---|---|---|
| | | Q-value (df); P-value | I-squared | |
| Addis Ababa | Z = 6.371, P = 0.000 | 1759.503 (4); P=0.000 | 99.773 | 0.000 |
| Amhara | Z = 4.790, P = 0.000 | 398.298 (5); P=0.000 | 98.745 | |
| Oromia | Z = 29.844, P = 0.000 | 100.358 (1); P=0.000 | 99.004 | |
| Tigray | Z =53.163, P = 0.000 | 3377.590 (9); P=0.000 | 99.734 | |
| Total between | - | 2481.082 (3); P=0.000 | - | - |

A random effects model was used to combine studies within each subgroup. The study-to-study variance (tau-squared) was assumed to be the same for all subgroups; this value is computed within subgroups and then pooled across subgroups.

**Fig 3. Forest plot for subgroup analysis of incidences of human exposure by regions** [7,16,19,24,27].

(95% CI: 46.29 to 195.69), followed by equines, 19.57 (95% CI: −1.85 to 40.98), and cattle, 18.08 (95% CI: 1 to 35.15). The pooled incidence rates of rabies were 13.03 (95% CI: −3.60 to 29.66) and 4.26 (95% CI: −0.43 to 8.96) per 100,000 in cats and goats, respectively. However, the overall true effect sizes could not be different from zero in cats (Z-value = 1.535, $P$ = 0.125), equines (Z-value = 1.791, $P$ = 0.173), and goats (Z-value = 1.779, $P$ = 0.075), as also indicated by the 95% CI of the pooled estimates in each. On the contrary, the true effect sizes were different

**Incidence rate of human exposure to suspected rabid animals with one study or dataset removed**

| Study name | Statistics with study removed | | | Rate (95% CI) with study removed |
|---|---|---|---|---|
| | Point | Lower limit | Upper limit | |
| Beyene et al., 2018 | 0.000313450 | 0.000295205 | 0.000331695 | |
| Ebuy et al., 2019 (2012 data) | 0.000312062 | 0.000294228 | 0.000329896 | |
| Ebuy et al., 2019 (2013 data) | 0.000311118 | 0.000293332 | 0.000328903 | |
| Ebuy et al., 2019 (2014 data) | 0.000301517 | 0.000283782 | 0.000319252 | |
| Ebuy et al., 2019 (2015 data) | 0.000302699 | 0.000284983 | 0.000320414 | |
| Ebuy et al., 2019 (2016 data) | 0.000305083 | 0.000287328 | 0.000322839 | |
| Gebru et al., 2019 | 0.000328470 | 0.000309878 | 0.000347062 | |
| Jemberu et al., 2013 | 0.000338571 | 0.000319975 | 0.000357167 | |
| Mesfin, 2022 (2015 data) | 0.000648389 | 0.000609733 | 0.000687045 | |
| Mesfin, 2022 (2016 data) | 0.000636094 | 0.000598489 | 0.000673700 | |
| Mesfin, 2022 (2017 data) | 0.000365105 | 0.000345782 | 0.000384428 | |
| Mesfin, 2022 (2018 data) | 0.000345914 | 0.000327229 | 0.000364598 | |
| Mesfin, 2022 (2019 data) | 0.000328248 | 0.000310044 | 0.000346452 | |
| Ramos et al., 2015 | 0.000308182 | 0.000290194 | 0.000326169 | |
| Teklu et al., 2017 (2012 data) | 0.000331755 | 0.000313109 | 0.000350402 | |
| Teklu et al., 2017 (2013 data) | 0.000317000 | 0.000298600 | 0.000335399 | |
| Teklu et al., 2017 (2014 data) | 0.000305752 | 0.000287579 | 0.000323924 | |
| Teklu et al., 2017 (2015 data) | 0.000311755 | 0.000293456 | 0.000330054 | |
| Yibrah and Damtie, 2015 (2011 data) | 0.000357905 | 0.000338873 | 0.000376937 | |
| Yibrah and Damtie, 2015 (2012 data) | 0.000361752 | 0.000342590 | 0.000380915 | |
| Yibrah and Damtie, 2015 (2013 data) | 0.000367214 | 0.000347801 | 0.000386627 | |
| Yizengaw et al., 2018 (2015/2016 data) | 0.000354063 | 0.000335171 | 0.000372954 | |
| Yizengaw et al., 2018 (2016/2017 data) | 0.000351877 | 0.000333063 | 0.000370691 | |
| | 0.000336512 | 0.000318150 | 0.000354874 | |

-0.01   -0.01   0.00   0.01   0.01

**Fig 4. Forest plot of sensitivity analysis for incidence rates of human exposure to rabid animals** [7,16,19,24,27].

from zero in dogs (Z-value = 3.175, $P = 0.002$) and cattle (Z-value = 2.071, $P < 0.001$). Considerable heterogeneities were found for the overall combined effect sizes in all the animals except for goats, where no variation was revealed among the included studies (Q-value (df) = 1.097 (1), $P = 0.295$; I-squared = 8.807) (Figs a, b, c, d, and e in S3 Information).

## Discussion

In the current review, the incidence rates of human exposures to suspected rabid animals from the incidence data of the included studies ranged from 0 to 422 (95% CI: 394 to 450) per 100,000 persons annually. This review revealed that estimates of human exposure to rabid animals were not uniform in the country, where the incidences varied widely across the regions as well as within the regions. However, the information can valuably alert the local or regional administrators to proactively plan for the prevention of rabies, such as purchasing optimal doses of post-exposure prophylaxis. The annual incidence rates of suspected human rabies from the datasets of the included studies ranged from 0 to 4.19 (95% CI: 2.61 to 5.77) per 100,000 humans. This finding varied from the systematic review results of Southeast Asia, which reported the incidence rates of rabies ranging between 0.1 and 117.2 per 100,000 persons [28]. Geographical differences might be the reason for the variations.

Among the animal studies included in the current review, the first and second highest records of incidence rates of rabies, 1880.19 [14] and 911.16 [18], were reported from cattle,

**Incidence rate of suspected human rabies**

| Study name | Statistics for each study | | | Rate and 95% CI |
|---|---|---|---|---|
| | Rate | Lower limit | Upper limit | |
| Beyene et al., 2018 | 0.000028037 | 0.000012174 | 0.000043901 | |
| Jemberu et al., 2013 | 0.000023411 | -0.000003081 | 0.000049902 | |
| Menghistu et al., 2018 | 0.000041251 | 0.000035885 | 0.000046617 | |
| Ebuy et al., 2019 (2012 data) | 0.000029437 | 0.000005883 | 0.000052991 | |
| Ebuy et al., 2019 (2013 data) | 0.000009780 | -0.000003774 | 0.000023334 | |
| Ebuy et al., 2019 (2014 data) | 0.000011142 | 0.000000223 | 0.000022062 | |
| Ebuy et al., 2019 (2015 data) | 0.000027527 | 0.000009543 | 0.000045511 | |
| Ebuy et al., 2019 (2016 data) | 0.000016826 | 0.000002078 | 0.000031574 | |
| Mesfin, 2022 (2015 data) | 0.000000129 | -0.000000229 | 0.000000487 | |
| Mesfin, 2022 (2016 data) | 0.000000124 | -0.000000219 | 0.000000467 | |
| Mesfin, 2022 (2017 data) | 0.000001186 | 0.000000146 | 0.000002225 | |
| Mesfin, 2022 (2018 data) | 0.000000227 | -0.000000218 | 0.000000673 | |
| Mesfin, 2022 (2019 data) | 0.000000436 | -0.000000168 | 0.000001039 | |
| Wakgari et al., 2022 | 0.000041925 | 0.000026111 | 0.000057739 | |
| Overall estimate | 0.000004570 | 0.000002930 | 0.000006210 | |

-0.01  -0.01  0.00  0.01  0.01

Heterogeneity: Q-value = 295.141 (df = 13), P-value = 0.000
I-squared = 95.595
Tau-squared = 0.000
Test of null: Z-value = 5.461, P-value = 0.000

**Fig 5. Forest plot for the incidence rates of suspected human rabies** [ 7,17,19,22,25,27].

while a relatively smaller incidence of rabies, 412.83 per 100,000, was reported from dogs [19]. These different figures between cattle and dogs might indicate that a single rabid dog can infect many cattle during the episode of the disease. In general, diagnosis of rabies based on the clinical presentation alone might overestimate the recorded incidence rates. This possible justification could be supported by one of the included studies where higher estimates of incidence rates were recorded in clinically suspected animals than in laboratory-confirmed cases, 92.8 versus 70.2 in dogs and 5.7 versus 3.2 in cats [26].

To generate the pooled incidence rates for rabies and related outcome conditions, studies eligible for meta-analysis were segregated and utilized according to the outcomes reported. Twenty-three incidence data sets from nine studies were utilized for estimating the overall incidence rate of human exposure to suspected rabid animals. The annual pooled incidence rate of human exposure to suspected rabid animals per 100,000 humans was 33.65 (95% CI: 31.82 to 35.49) in the current meta-analysis. This result is much smaller than the pooled incidence rate of animal bites reported from Iran, 13.20 (95% CI: 12.10 to 14.30) per 1000 population [29]. The differences might be due to geographical variation, under-reporting from Ethiopia, and/or differences in the definitions of rabies exposures. Higher incidence rates of canine rabies exposures are also reported from developed and developing regions of the world: Eastern Europe (120.9 from Croatia, 188.77 from Serbia, and 157.98 from Ukraine) and North Africa (277.15 from Algeria and 185.71 from Morocco) per 100,000 populations

## Incidence rate of suspected human rabies with one study or dataset removed

| Study name | Statistics with study removed | | | Rate (95% CI) with study removed |
| --- | --- | --- | --- | --- |
| | Point | Lower limit | Upper limit | |
| Beyene et al., 2018 | 0.000004236 | 0.000002616 | 0.000005857 | |
| Jemberu et al., 2013 | 0.000004481 | 0.000002843 | 0.000006120 | |
| Menghistu et al., 2018 | 0.000000840 | 0.000000005 | 0.000001674 | |
| Ebuy et al., 2019 (2012 data) | 0.000004408 | 0.000002776 | 0.000006039 | |
| Ebuy et al., 2019 (2013 data) | 0.000004491 | 0.000002842 | 0.000006140 | |
| Ebuy et al., 2019 (2014 data) | 0.000004411 | 0.000002762 | 0.000006061 | |
| Ebuy et al., 2019 (2015 data) | 0.000004317 | 0.000002690 | 0.000005943 | |
| Ebuy et al., 2019 (2016 data) | 0.000004393 | 0.000002753 | 0.000006032 | |
| Mesfin, 2022 (2015 data) | 0.000007129 | 0.000004909 | 0.000009349 | |
| Mesfin, 2022 (2016 data) | 0.000007210 | 0.000004969 | 0.000009451 | |
| Mesfin, 2022 (2017 data) | 0.000005413 | 0.000003575 | 0.000007251 | |
| Mesfin, 2022 (2018 data) | 0.000006609 | 0.000004520 | 0.000008698 | |
| Mesfin, 2022 (2019 data) | 0.000006040 | 0.000004085 | 0.000007995 | |
| Wakgari et al., 2022 | 0.000003968 | 0.000002386 | 0.000005549 | |
| | 0.000004570 | 0.000002930 | 0.000006210 | |

-0.01  -0.01  0.00  0.01  0.01

**Fig 6. Forest plot for sensitivity analysis of incidence rate of suspected human rabies** [ 7,17,19,22,25,27].

[30]. The variation might be due to a systematic article search in our case, but data with higher figures might be preferably presented in others.

The reliability of the meta-analysis result was demonstrated by the sensitivity analysis results for the incidence of human exposure to rabid animals, which were consistent with the overall effect size when most of the incidence data were eliminated one at a time. In a few cases, however, the overall results were noticeably changed when two datasets from the study reported by Mesfin [27] were omitted one by one, giving the pooled incidence rates of 64.84 (95% CI: 60.97 to 68.70) and 63.61 (95% CI: 59.85 to 67.37) per 100,000 populations, indicating that these incidence data were extremely small compared with others. The Egger's regression test was performed to check for publication bias and obvious publication bias was detected among the publications that reported the incidence of human exposure to suspected rabid animals (Intercept = 22.06, 95% CI: 18.13 to 26.00; P < 0.001). This result might show that the missing studies differ systematically from the observed studies.

It is important to note that in rabies-endemic countries, trustful reports of incidence rates of rabies and rabies exposure are lacking, especially in developing countries. So far, it has been described that official reports underestimated the true number of human rabies cases as well as their exposures [4,31]. In this manner, the Ethiopian national annual estimate from official reports indicated exposure cases of 12 per 100,000 populations [32]. The current meta-analysis reported an annual pooled incidence rate of 33.65 (95% CI: 31.82 to 35.49) human exposures per 100,000 populations, which is almost three times the one previously officially reported.

In the current meta-analysis of human rabies exposures, considerable heterogeneity was recognized for the pooled estimate. From this, we could understand that variations existed in the effect sizes between the individual studies (Q (df) = 11222.628 (22) with $P < 0.001$). The presence of heterogeneity among studies could also be explained by the heterogeneity statistics, I-squared ($I^2$), which indicated that about 99.804% of the variability in the effect-size estimates was due to the actual differences between studies. A further subgroup analysis was performed by regions to explore whether regions could be the source of heterogeneity between the individual studies included in determining the incidence of human exposure to rabid animals. Substantial heterogeneities were also seen from the subgroup analysis between regions (Q (df) = 2481.082 (3); P < 0.001), indicating that regions of the studies would be sources of heterogeneity. Regionally, the pooled incidence rates of human exposure to suspected rabid animals ranged from 5.52 (95% CI: 3.26 to 7.79) to 138.91 (95% CI: 129.79 to 148.03) per 100,000 persons. The highest pooled incidence rate was recorded in Oromia, 138.91 (95% CI: 129.79 to 148.03), followed by Tigray, 83.32 (95% CI: 80.25 to 86.39), while the smallest estimate was from Amhara, 5.52 (95% CI: 3.26 to 7.79), followed by Addis Ababa, 7.58 (95% CI: 5.25 to 9.91). The reasons for the differences among the regions could be due to the following: individuals who are exposed to rabies often prefer to see traditional healers for the diagnosis and treatment of the disease because of their cultural background, lack of knowledge, or limited accessibility to medical treatment.

In determining the pooled incidence rate of suspected human rabies, six studies with 14 datasets were included, providing an effect size of 4.57 (95% CI: 2.93 to 6.21) per one million humans. This finding was significantly higher than the incidence of human rabies in China, 0.86 per 1,000,000 population [33]. The difference might be due to the extensive implementation of surveillance systems to determine the real situation of rabies as well as the extensive dog vaccination in China [33], in contrast to the low coverage of dog vaccination [6,9] and lack of active surveillance in Ethiopia. Our result was also higher than the report from sub-Saharan Africa with 1.4 cases (95% CI: 0.5–4.0) per 1,000,000 populations and lower than the one from South Asia with 9.2 cases (95% CI: 4.7–18.1) per 1,000,000 populations [33]. Geographical variations might be the reason for the differences.

For the overall pooled incidence of suspected human rabies, sensitivity analysis was undertaken, and one study reported by Menghistu and others found an outlier [25]. When the outlier was removed, the combined effect of the rest of the studies/datasets gave a very small pooled incidence, < 1 per one million. However, the overall results were not noticeably changed by omitting each of the other studies/datasets. Among the publications included for determining the incidence of human rabies, it was shown that the missing studies differed systematically, as confirmed by Egger's regression test (Intercept = 4.01505, 95% CI: 1.43272 to 6.59738; P = 0.0027).

In the present review, five or fewer studies were included to determine the pooled incidence rates of suspected rabies in five different types of animals: 5 studies for cattle, 4 for dogs, 3 for equines, and 2 studies each for cats and goats. The highest pooled incidence was recorded in dogs, 120.99 (95% CI: 46.29 to 195.69), followed by equines, 19.57 (−1.85 to 40.98), and then cattle, with 18.08 (95% CI: 1 to 35.15) per 100,000 populations. In Ethiopia, rabies is a potential problem for domestic animals such as cattle and others, as many households own dogs, usually for herding livestock or guarding property. This is because dogs are kept in close contact with cattle and other animals, transmitting the virus to them through a bite. By affecting cattle, rabies may have extensive economic impacts at the household and country levels, in addition to the effects on human health [34].

Regarding the associated factors of exposure to rabid animals, some reports included in the review described the relationship between certain socio-demographic features and

the occurrence of rabies or its exposures. It was revealed that males were more exposed to rabid animals [20,24]. This could be due to the working environment of males, where they usually stay outside the home where they encounter stray dogs, or because male children have the habit of playing outside their homes with their age mates. Compared with adults, children <15 years old were more likely to be exposed to rabid animals [15,16], and most human rabies deaths occurred among children [25]. This might be explained by the fact that children have an interest in playing with pets, such as dogs and cats because they have poor awareness about rabies and its consequences. On the contrary, one study reported that adults were more affected [24]. Contradictory reports were also reported regarding the residence and incidence of rabies cases. Two studies reported that rural inhabitants were more prone to exposure to rabid animals [15,19]. Another study reported that urban residents were more exposed [20]. Such discrepancies require more explanations through further investigations.

A limited number of studies reported what measures were taken once humans were exposed to suspected rabid animals or the status of vaccination of dogs/animals inflicting bites. Almost 99% of individuals exposed to rabid animals had completed the required doses of nerve tissue vaccines [16]. On the other hand, all the dogs inflicting unprovoked bites were unvaccinated [24]. These reports indicate that the absence or low coverage of dogs' vaccination is one of the major problems that increase the risk of rabies and the costs associated with post-exposure treatments.

In the current systematic review and meta-analysis, some limitations were noted. Among them, one is that we found substantial heterogeneity in the estimation of the incidence rates of our outcome conditions across the included studies. In this regard, regions were identified as one of the sources of heterogeneity for the incidence of rabies exposures in humans. However, the other possible moderators for the heterogeneities have not been identified because of the difficulty of using such variables. The majority of the included studies were retrospective, where diagnosis bias could have occurred. The other problem was that the amount of data available per region was variable, even with no reports from the majority of the regions. In most of the papers included in the review, a single published article in a given publication year has a number of datasets of incidence rates due to the differences in person-years. In this aspect, we couldn't analyze the trends of incidence rates by publication year. Hence, the results from the review have to be cautiously interpreted.

Despite the above-highlighted limitations, this review and meta-analysis have some strength. It used a predefined and registered protocol; this may ensure the reliability and scientific nature of the study. Many databases of medical literature were exhaustively searched for this review, which helped to retrieve a possible large number of studies available over long publication years. Furthermore, the search for published studies was supported by free online search engines. In all the phases of the review process, three independent investigators were also involved. In the original studies included in the review, sufficient data existed to answer our objectives.

## Conclusions and Recommendations

The current study showed a high pooled incidence rate of human exposure to rabid animals. The overall incidence rates of rabies in humans and animals were also alarming. As rabies is one of the neglected zoonotic diseases, research in this area is limited in Ethiopia calling for further researches from different corners of the country. In general, the summarized report from our study provided important pictures of the risk of rabies to the health of public, alarming that attention should be paid to the prevention and control of rabies by the Ethiopian Ministry of Health in collaboration with the country's animal health sectors. As the high

number of unvaccinated dogs is a major factor in the incidence of rabies and its associated public impacts, increasing the vaccination coverage of dogs is mandatory. This can be done through canine vaccination campaigns that can boost herd immunity and reduce the risk of human rabies exposure, necessitating strong governmental concerns and persistent community mobilization by both human and animal health sectors. Overall, integrating rabies prevention and control with other disease programs and strengthening intersectoral and transdisciplinary collaboration through one health approach is required.

## Supporting information

**S1 Information.  Search terms and strings in different databases** .
(DOCX)

**S2 Information.  Extracted datasets of the included studies.**
(XLSX)

**S3 Information.  Forest plots for the incidence rates of rabies in different animals.**
(PDF)

## Author contributions

**Conceptualization:** Belay Tafa Regassa, Andargachew Mulu, Gadissa Bedada Hundie.

**Data curation:** Belay Tafa Regassa, Wagi Tosisa, Daniel Eshetu, Andargachew Mulu, Gadissa Bedada Hundie.

**Formal analysis:** Belay Tafa Regassa.

**Investigation:** Belay Tafa Regassa.

**Methodology:** Belay Tafa Regassa, Wagi Tosisa, Daniel Eshetu, Andargachew Mulu, Gadissa Bedada Hundie.

**Project administration:** Belay Tafa Regassa.

**Software:** Belay Tafa Regassa.

**Supervision:** Andargachew Mulu, Gadissa Bedada Hundie.

**Validation:** Belay Tafa Regassa, Wagi Tosisa, Daniel Eshetu, Andargachew Mulu, Gadissa Bedada Hundie.

**Writing – original draft:** Belay Tafa Regassa.

**Writing – review & editing:** Andargachew Mulu, Gadissa Bedada Hundie.

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
