## [Decision Letter · Decision Letter 0]

17 Jun 2024

Dear Mr. Regassa,

Thank you very much for submitting your manuscript "Incidence, risk factors, and control of Rabies in Ethiopia: A systematic review and meta-analysis" for consideration at PLOS Neglected Tropical Diseases. As with all papers reviewed by the journal, your manuscript was reviewed by members of the editorial board and by several independent reviewers. In light of the reviews (below this email), we would like to invite the resubmission of a significantly-revised version that takes into account the reviewers' comments.

We cannot make any decision about publication until we have seen the revised manuscript and your response to the reviewers' comments. Your revised manuscript is also likely to be sent to reviewers for further evaluation.

Sincerely,

Sergio Recuenco

Academic Editor

David Safronetz

Section Editor

Reviewer's Responses to Questions

**Key Review Criteria Required for Acceptance?**

**Methods**

-Are the objectives of the study clearly articulated with a clear testable hypothesis stated?

-Is the study design appropriate to address the stated objectives?

-Is the population clearly described and appropriate for the hypothesis being tested?

-Is the sample size sufficient to ensure adequate power to address the hypothesis being tested?

-Were correct statistical analysis used to support conclusions?

-Are there concerns about ethical or regulatory requirements being met?

Reviewer #1: (No Response)

Reviewer #2: the objectives of the study were clearly articulated and clear

Yes, the study design, systematic review is appropriate and rigorously followed to address the objective

The population was clearly described and suitable for testing

The selected articles used for the study was based on the selection criteria, which for this study was clearly selected and was significant to support the results produced.

Yes, the statistical analysis used was correct even though poorly presented and it will be important for the author to present in a simplified manner.

No concerns on ethics or regulatory requirement as the authors registered the selected articles in PROSPERO with number CRD4202346879

**Results**

-Does the analysis presented match the analysis plan?

-Are the results clearly and completely presented?

-Are the figures (Tables, Images) of sufficient quality for clarity?

Reviewer #1: (No Response)

Reviewer #2: Yes it does, the analysis was able to put data from different sources and and different years and calculate the incidence rate. the weakness however, is that the differences in the years might affect the results produced such that the current rate might not be real due to the cumulative approach. However, it responded to the research question and is still informative.

The results for risk factors were not clearly presented, however, the paragraph is loaded with lots of information which could be clearly presented. For example

Sex and age are associated factors. They should be presented and discussed in different paragraphs.

Also, the author could explore some other factors associated to rabies exposure to improve on the qualitative data.

The figures and tables created ok except for those of incidence rates which are different and should be harmonized.

**Conclusions**

-Are the conclusions supported by the data presented?

-Are the limitations of analysis clearly described?

-Do the authors discuss how these data can be helpful to advance our understanding of the topic under study?

-Is public health relevance addressed?

Reviewer #1: (No Response)

Reviewer #2: The conclusion is supported by data but could be improved upon by including some more information as per the objective and expand a little on the recommendation and possibly highlighting to whom the study is addressed to or services concerned.

The limitation are not defined and should be included in the study.

yes , they showed evidence of high incidences and showed different rabid animals. However, the authors should mention the opportunities for research in this domain based on this study.

Public health relevance is not directly addressed. The authors addressed health in general .

**Editorial and Data Presentation Modifications?**

Reviewer #1: (No Response)

Reviewer #2: The article needs minor revisions, such as grammatical errors, harmonization of the tables .

Also, the author should describe the data in terms of years. For example, how many articles were collected in 2013 an how many in 2011. This will help the reader not only understand possible trends, but will give some reliability to the data. Also, the authors should state the limitation of the data analysis. One of them being that, it is not stated the source of the data, some of the studies might be using same data etc.. All of that is not very clear in the study.

For the qualitative data, some description of the method used should be mentioned and the presentation improved upon. The author could also ensure that just sex and age are the only human exposure factor to rabiesThe .

Apart from that minor revision

**Summary and General Comments**

Reviewer #1: Regassa et al. conducted a systematic review and meta-analysis on the incidence, risk factors, and control of rabies in Ethiopia. They reported a high pooled incidence rate of human exposure to rabid animals and significant overall incidence rates of rabies in both humans and animals. The authors emphasized the need for strengthening intersectoral and transdisciplinary collaborations using a One Health approach to enhance rabies prevention and control efforts.

However, the introduction of the study is currently too lengthy. To improve the introduction, the authors should consider providing a concise background that references existing studies, particularly other systematic review and meta-analyses from Ethiopia ( see some below) . This will help justify the importance and relevance of their study within the existing body of research on rabies in Ethiopia. ( Belete S, Meseret M, Dejene H, Assefa A. Prevalence of dog-mediated rabies in Ethiopia: a systematic review and Meta-analysis from 2010 to 2020. One Health Outlook. 2021 Aug 4;3(1):16. doi: 10.1186/s42522-021-00046-7. PMID: 34344487; PMCID: PMC8336362.; Gelgie AE, Cavalerie L, Kaba M, Asrat D, Mor SM. Rabies research in Ethiopia: A systematic review. One Health. 2022 Oct 18;15:100450. doi: 10.1016/j.onehlt.2022.100450. PMID: 36532671; PMCID: PMC9754932; Aklilu M, Tadele W, Alemu A, Abdela S, Getahun G, Hailemariam A, Tadesse Y, Kitila G, Birhanu E, Fli I, Getachew A, Mulugeta Y. Situation of Rabies in Ethiopia: A Five-Year Retrospective Study of Human Rabies in Addis Ababa and the Surrounding Regions. J Trop Med. 2021 Feb 19;2021:6662073. doi: 10.1155/2021/6662073. PMID: 33679992; PMCID: PMC7910060.)

The discussion section is lengthy and contains several repetitions of the results. Previous studies conducted in Ethiopia and elsewhere have emphasised many of the discussion points. Therefore, the authors should focus on synthesizing key findings and providing new insights or interpretations. They should avoid repetition of results and instead highlight novel contributions or implications of their study. Additionally, the authors could compare their findings with previous studies from Ethiopia and other regions to underscore similarities, differences, or advancements in knowledge.

Reviewer #2: Strength - very important but neglected diseases

- used both qualitative and quantitative analysis

- clear inclusion and exclusion criteria

opportunity for more research on rabies across SSA

Weakness- No year limit for articles

- comparison of results with other countries in the discussion is not very visible.

The study is not completely novel however, opens widows for rigorous data collection on rabies and its associated risk factors.

Generally, the study is very interesting and fairly exploited especially at the level of the qualitative study.

PLOS authors have the option to publish the peer review history of their article (what does this mean? ). If published, this will include your full peer review and any attached files.

**Do you want your identity to be public for this peer review?** For information about this choice, including consent withdrawal, please see our Privacy Policy .

Reviewer #1: No

Reviewer #2: No
---

## [Decision Letter · Decision Letter 1]

11 Dec 2024

PNTD-D-24-00387R1

Incidence, risk factors, and control of Rabies in Ethiopia: A systematic review and meta-analysis

Dear Dr. Regassa,

Thank you for submitting your manuscript to PLOS Neglected Tropical Diseases. After careful consideration, we feel that it has merit but does not fully meet PLOS Neglected Tropical Diseases's publication criteria as it currently stands. Therefore, we invite you to submit a revised version of the manuscript that addresses the points raised during the review process.

Please submit your revised manuscript within 60 days Jan 10 2025 11:59PM. If you will need more time than this to complete your revisions, please reply to this message or contact the journal office at plosntds@plos.org. Please include the following items when submitting your revised manuscript:

We look forward to receiving your revised manuscript.

Kind regards,

Sergio Recuenco

Academic Editor

David Safronetz

Section Editor

Shaden Kamhawi

co-Editor-in-Chief

Paul Brindley

co-Editor-in-Chief

**Additional Editor Comments:**

The manuscript approaches important aspect for rabies in Ethiopia, that deserves publication, therephore I encourage the authors to work as needed to fully absolve the remaining points from the reviewer. I look forward to receive the improved revised version.

**Journal Requirements:**

Please upload all main figures as separate Figure files in .tif or .eps format. For more information about how to convert and format your figure files please see our guidelines:

**Reviewers' Comments:**

Reviewer's Responses to Questions

**Key Review Criteria Required for Acceptance?**

**Methods**

-Are the objectives of the study clearly articulated with a clear testable hypothesis stated?

-Is the study design appropriate to address the stated objectives?

-Is the population clearly described and appropriate for the hypothesis being tested?

-Is the sample size sufficient to ensure adequate power to address the hypothesis being tested?

-Were correct statistical analysis used to support conclusions?

-Are there concerns about ethical or regulatory requirements being met?

Reviewer #1: While the authors have made some efforts to address the concern raised, they failed to justify why this investigation was conducted, including how it might support the country's rabies prevention and control efforts. Clearly, Several studies have been done in Ethiopia that have quantified risk and provided helpful information (https://onehealthoutlook.biomedcentral.com/articles/10.1186/s42522-021-00046-7). Including the ones the authors reported . According to the CDC, rabies results in the death of 2,700 persons annually in Ethiopia, making it one of the highest death rates in the world (7). Furthermore, extrapolation of the district data to the national level indicated an annual estimate of approximately 3,000 human deaths, 194,000 DALYs, and 97,000 exposed persons, requiring on average 2 million USD in treatment costs per year countrywide (8).

The authors' justification for conducting this study is not clear (119- 121) and cannot in any way support the planning of mass dog vaccination, as nowhere in the paper did the authors report maps of locations needing targeted intervention.

Lines 119 to 121. However, the overall risk of rabies in a population has not yet been estimated in the country. Information on the rabies incidence rate is important for notifying the population at risk of the disease and for planning effective prevention, control, and elimination strategies.

The manuscript is too lengthy and contains a lot of repetition(Lines 95 and 106 say the same thing) and contractions. For example, according to the authors, there is limited capacity for testing (Line 92); when you go further into Line 107, the authors reported 70.3% lab confirmation. The authors of this paper need to justify why this study was conducted by articulating findings from previous studies in the country and how their work will advance rabies prevention and control efforts in the country.

**Results**

-Does the analysis presented match the analysis plan?

-Are the results clearly and completely presented?

-Are the figures (Tables, Images) of sufficient quality for clarity?

Reviewer #1: (No Response)

**Conclusions**

-Are the conclusions supported by the data presented?

-Are the limitations of analysis clearly described?

-Do the authors discuss how these data can be helpful to advance our understanding of the topic under study?

-Is public health relevance addressed?

Reviewer #1: (No Response)

**Editorial and Data Presentation Modifications?**

Reviewer #1: (No Response)

**Summary and General Comments**

Reviewer #1: (No Response)

PLOS authors have the option to publish the peer review history of their article (what does this mean? ). If published, this will include your full peer review and any attached files.

**Do you want your identity to be public for this peer review?** For information about this choice, including consent withdrawal, please see our Privacy Policy .

Reviewer #1: No

**Figure resubmission:**

**Reproducibility:**



---

## [Editor Report · Decision Letter 2]

28 Jan 2025

Dear Mr. Regassa,

We are pleased to inform you that your manuscript 'Incidence, risk factors, and control of Rabies in Ethiopia: A systematic review and meta-analysis' has been provisionally accepted for publication in PLOS Neglected Tropical Diseases.

Best regards,

David Safronetz, Ph.D.

Section Editor

David Safronetz

Section Editor

Shaden Kamhawi

co-Editor-in-Chief

Paul Brindley

co-Editor-in-Chief

---

## [Editor Report · Acceptance letter]

Dear Mr. Regassa,

We are delighted to inform you that your manuscript, "Incidence, risk factors, and control of Rabies in Ethiopia: A systematic review and meta-analysis," has been formally accepted for publication in PLOS Neglected Tropical Diseases.

Best regards,

Shaden Kamhawi

co-Editor-in-Chief

Paul Brindley

co-Editor-in-Chief
